# Development of In-Process Temperature Measurement of Grinding Surface with an Infrared Thermometer

**Yukio Ito \*, Yoshiyuki Kita, Yoshiya Fukuhara, Mamoru Nomura and Hiroyuki Sasahara** 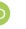

Department of Mechanical Systems Engineering, Tokyo University of Agriculture and Technology, Tokyo 184-8588, Japan; yoshiyukikt41@gmail.com (Y.K.); josiah@mh1.117.ne.jp (Y.F.); mamoru.nomura@ibarakiseito.com (M.N.); sasahara@cc.tuat.ac.jp (H.S.)
**\*** Correspondence: jms007@me.ccnw.ne.jp; Tel.: +81-42-388-7417

**Abstract:** Heat generation is a critical issue in grinding. If the grinding point generates significant heat, dimensional and shape accuracy may decrease due to thermal deformation, and the machined surface may deteriorate due to grinding burn. Therefore, monitoring the temperature during grinding is important to obtain ideal machining results. In this research, we develop a new method to measure the grinding surface and grinding wheel surface temperature during in-process machining. The proposed method measures the temperature of the grinding surface through small holes in a rotating grinding wheel. Using this method, we measured the temperature of the grinding surface during the dry grinding of carbon fiber reinforced plastics (CFRP). Temperature of the grinding surface was measured every 1/4 rotation of the grinding wheel at any depth of cut, assuming precision grinding, rough grinding, and high-efficiency grinding. The measurement value changed depending on the temperature measurement position of the infrared thermometer from numerical analysis of the grinding surface temperature. We also found that when the cut depth was small, the temperature, including the surface of the workpiece before machining, was measured at a specific temperature measurement position. The newly developed temperature measurement method was capable of in-process measurement of the grinding surface temperature and of detecting temperature rise when the grinding wheel was clogged.

**Keywords:** grinding; temperature measurement; in-process monitoring; CFRP; infrared thermometer

## 1. Introduction

Grinding is employed for a wide range of finishing processes because it can easily achieve higher dimensional accuracy and fine surface roughness for high-hardness materials. However, the grinding wheel surface changes during the process due to the self-sharpening effect, in which the wear and drop off of the abrasive grains occur appropriately. Even in an electroplated grinding wheel, the state of the grinding surface changes due to wear and clogging of the abrasives. The setting and management of the grinding conditions largely depend on the skill of workers. In addition, it should be noted that a large amount of heat generated during grinding tends to lead to the deterioration of dimensional and shape accuracy due to thermal deformation and grinding burn, which can change the material properties of the work material.

Grinding burns sometimes cannot be detected from appearance only after grinding, and management during the process is important [1]. Grinding is effective not only for metallic materials but also for fiber-reinforced composites, such as carbon fiber reinforced plastics (CFRP). If the temperature exceeds the glass transition point, however, the deterioration of the mechanical properties of materials can be expected. It is therefore necessary to know the grinding point temperature.

In recent years, the manufacturing phase has shifted from an era that relied on hardware technology to one that manufactures fabricated products at low cost with a short lead

time by the effective use of various information sources. One of the most important issues is to collect machining process-related information in-process, to extract valid data from obtained big data, analyze them, and feed them back to the manufacturing process. In recent research, a method has been used to acquire data related to processing at the same time using multiple sensing units and use them for monitoring the machining status [2–4], and a method of monitoring in real time by transmitting the obtained machining data wirelessly has also been used [5]. In addition, a method for identifying the wear of the grinding wheel [6] and predicting the surface roughness [7] using machine learning has also been reported.

In terms of in-process grinding monitoring, clogging detection using an eddy current [1] and in-process discrimination of abrasive grain blister using a lightning tube [8] have been reported. They have not been put to practical use, however, because shedding, clogging and grazing occur in a complex and continuous manner in the actual grinding process. In addition, the surface roughness monitoring system [9] and in-process discrimination of the grinding wheel surface state using a laser displacement sensor [10–13] has been reported. However, heat generation, an important problem in grinding, has not been evaluated.

Regarding heat generation in grinding, observing the temperature of the surface of the workpiece material [14–18] and the elastic wave (AE wave) generated during machining [19,20] have been reported.

Some of the researchers involved in this study developed a method for monitoring the surface temperature of the grinding wheel by embedding a thermocouple in the wheel, as shown in Figure 1A [21]. They demonstrated that it is possible to continuously discriminate the grinding wheel surface during machining [22,23]. Against these backgrounds, we first propose a new temperature-monitoring method for the grinding surface using an infrared radiation sensor. The sensing device itself has been known for a long time, but it is here used in a novel manner to monitor infrared information in-process utilizing small holes in a thin-walled hollow grinding wheel (hereafter, HGW). In this study, we first clarify the basic characteristics of the proposed temperature measurement method. We then demonstrate that the proposed method is effective for in-process temperature measurement of the grinding surface and the detection of machining abnormalities when applied to CFRP dry grinding.

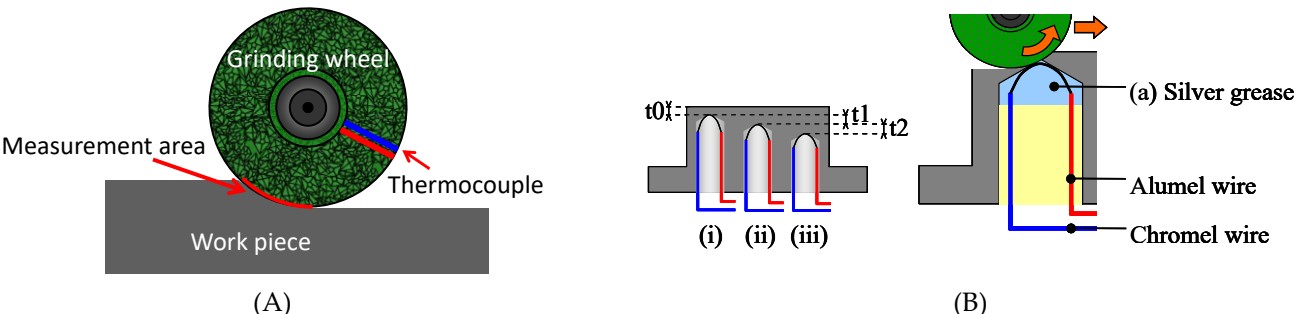

**Figure 1.** Conventional temperature measurement with the thermocouple. (**A**) Embedded in grinding wheel type [21]. (**B**) Embedded in workpiece type [24].

## 2. A Novel Concept for Grinding Surface Temperature Measurement with an Infrared Thermometer

Figure 2 shows the setup and measurement principle of the system for measuring the temperature of the grinding surface using the proposed infrared thermometer. The setup uses a thin-wall hollow electroplated grinding wheel, 84 mm in diameter. Although the grinding wheel is a straight-cup type, we consider it a grinding wheel because we use the side surface rather than the end surface. It is assumed that the measuring instrument will be embedded in the hollow part in the future. In addition, a thin wall structure can be

fabricated by either a machining process or sheet metal forming process, enabling low-cost fabrication of the base metal of the grinding wheel. A small hole was created on the thin-wall HGW, perpendicular to the tool axis, and then the workpiece, grinding wheel, and infrared thermometer were arranged in that order. The temperature on the grinding surface could then be measured through the two holes on the grinding wheel. Alternatively, the temperature on the HGW could also be measured when the outer surface of the grinding wheel was placed within a temperature measuring area of the infrared thermometer. Since the emissivity differs between the workpiece and the surface of the grinding wheel, it is necessary to convert the output value by its emissivity to obtain the correct temperature of the grinding wheel surface.

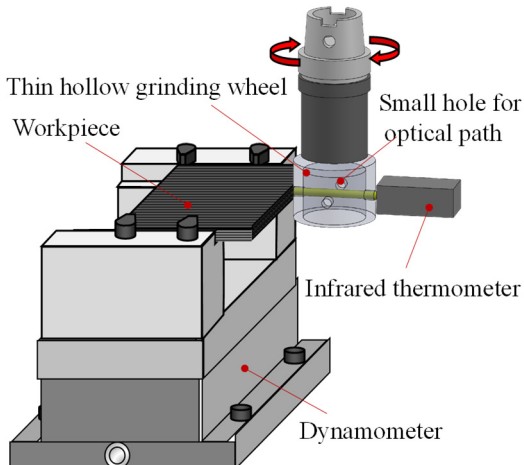

(**a**) Proposed temperature measurement setup with an infrared thermometer

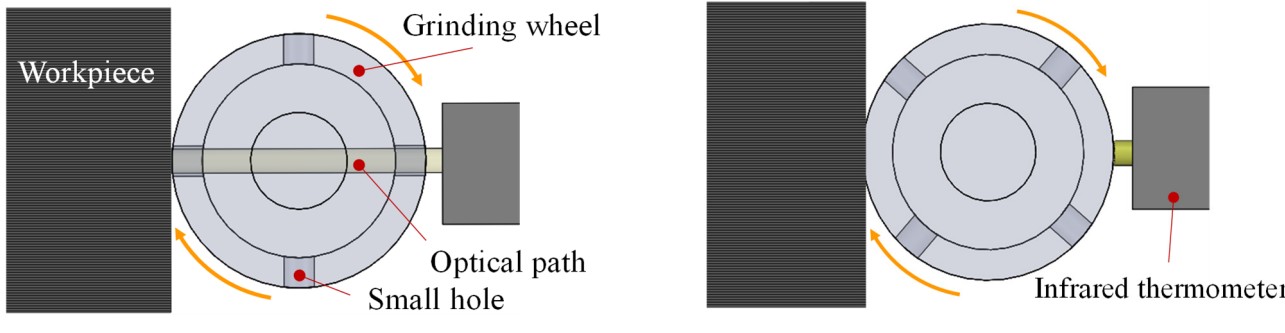

(**b**) Measurement on grinding surface       (**c**) Measurement on grinding wheel

**Figure 2.** Measurement concept and its setup.

The thermocouple method and infrared thermometer method are the conventionally reported temperature measurement methods. When using a thermocouple, it is necessary to embed the thermocouple in the grinding wheel or workpiece, as shown in Figure 1A,B [21,24], respectively. In general, both time resolution and spatial resolution can be high, which is very useful for research. However, assuming data collection on a production line, it is not realistic to embed thermocouples in all processing targets. There are also problems in putting this method to practical use when embedding the thermocouples in a grinding wheel.

The infrared thermometer can measure the temperature by detecting the infrared rays on the exposed part of the surface. If we use a thermography camera, the temperature distribution of the "visible part" can be easily measured. On the other hand, the "invisible parts", such as the contact area between grinding wheel and workpiece, cannot be measured. There are examples of measuring the internal temperature through an optical fiber, but this would not be realistic to install on the workpiece side on a production line.

In contrast, it is possible to measure the temperature of the contact arc region or grinding surface immediately after machining through small holes provided on the thin-walled HGW in the method we propose, even though the duration of the rotation of the grinding wheel is very short. Although it is necessary to create small holes in the grinding wheel, it is assumed that they will have little effect on the workpiece. Regarding the thin-walled HGW, we note that the measuring instrument, amplifier, transmitter, etc., can be miniaturized and placed in the hollow area in the future, and a monitoring grinding wheel could be realized.

## 3. Verification of Temperature Measurement Operation

### 3.1. Experimental Conditions

Here, to verify the basic operating characteristics of the proposed method, we investigated the responsiveness of the measured temperature waveform against the spindle rotation speed, targeting a blackbody furnace heated to 300 °C. Temperature measurement for a blackbody furnace differs from that for the grinding surface, the target temperature is maintained at a set temperature and CFRP chips do not interfere with the measurement. This allowed us to verify whether the temperature could be measured correctly through the two small holes of the rotating grinding wheel. The specifications of the infrared thermometer were as follows: measurement temperature range, 30–500 °C; measurement wavelength, 2.0 µm to 6.8 µm; measurement distance, 100 mm; temperature measurement target size, $\phi$8 mm; response time, 100 µs.

Since the temperature measurement target size is a $\phi$8 mm circle, the temperature of the target could be measured correctly when the infrared rays from the uniform temperature object were incident on the entire temperature-measurement target size. When the infrared rays from the target actually enter the sensor through the rotating holes, the output from the sensor depends on the opening between the target and the sensor. That is, as shown in Figure 3, it is determined by the degree of overlap of the two small holes on the workpiece side and sensor side provided on the grinding wheel. In the figure, the pink circles indicate the two small holes, and the blue dashed circle the temperature-measurement region. The hole on the sensor side is larger, as the distance to the sensor is small in this figure. Here, we ignore the effect of the thickness of the grinding wheel. When the two small holes start to overlap, the sensor begins to detect the input infrared ray (Figure 3b). Figure 3c shows the 100% incident state, and then the incident amount decreases in Figure 3d,e. Therefore, the size of the small hole relates to the time when the incident state was 100%. Thus, the experiment was conducted by changing the size and shape of the small holes on the grinding wheel. Figure 4 shows the two types of grinding wheels prepared with small measurement holes: Type A had four holes with a diameter of $\phi$10 mm at 90 degree intervals; Type B had two holes with a diameter of $\phi$18 mm at 180 degree intervals, and two oval holes with a major axis of 18 mm at 180 degrees.

Figure 5 presents a schematic diagram of the positional relationship between the optical path and grinding wheel at the moment the optical path of the infrared thermometer passes through the center of the small hole. The opening time $t_0$, at which point the opening area is larger than the temperature measurement target size, can be calculated by Equation (1). Here, the opening angle (i.e., the rotation angle of the grinding wheel from the position in Figure 5 until a part of the optical path is chipped) is $\theta$ (degrees), and the spindle speed is $R$ [r/min]. As shown in Figure 3b,d, there is a time input into the sensor before and after the $t_0$, so the temperature change is steeper than when inputting into the sensor from the hole on the workpiece side only. Table 1 lists the calculation results for the opening time $t_0$ [µs] for each grinding wheel and spindle speed, and Table 2 summarizes the experimental conditions. The experiment was carried out by changing the spindle speed as follows: 1000 r/min—spindle speed, where the opening time $t_0$ was sufficient for all grinding wheels for the response time of 100 µs of the infrared thermometer; 4500 r/min—spindle speed within the recommended range for electroplated grinding

wheel with a diameter of 84 mm; 7000 r/min—in the case of Type A, $t_0$ was 58.8 µs, less than the response time.

$$t_0 = \frac{2 \times \theta \times 60 \times 10^6}{360 \times R} \tag{1}$$

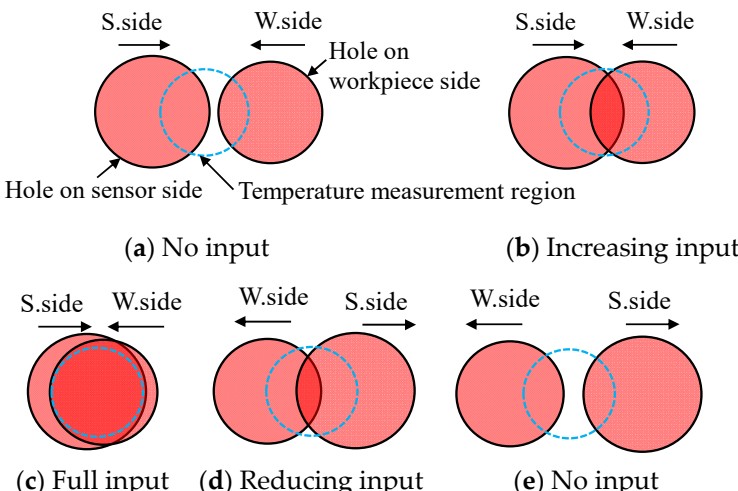

**Figure 3.** Transition of input to infrared sensor through two holes on the grinding wheel.

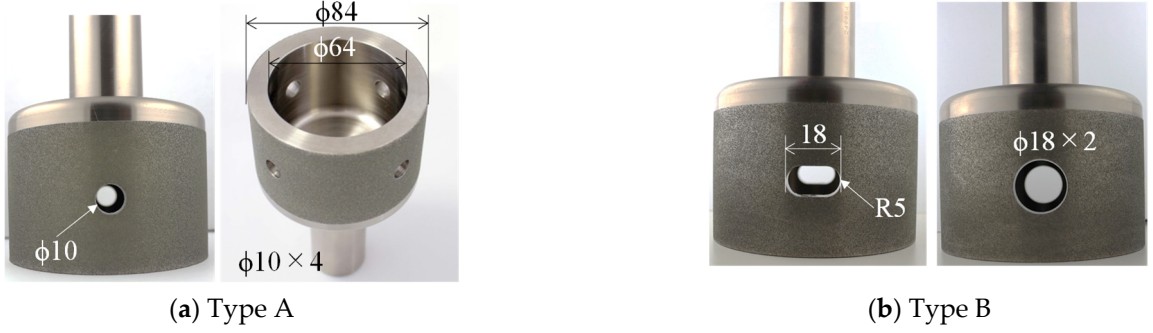

(**a**) Type A                    (**b**) Type B

**Figure 4.** Configuration of grinding wheel and holes for optical path.

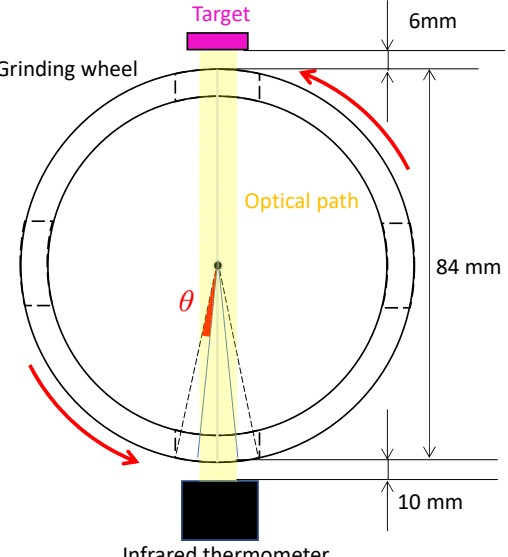

**Figure 5.** Aperture angle of grinding wheel (Type B).

**Table 1.** Opening time, $t_0$, for each grinding wheel and spindle speed.

| Grinding wheel | Type A ($\varphi$10) | Type B($\varphi$18) |
|---|---|---|
| Aperture angle $\theta$ [deg] | 1.24 | 6.77 |
| Spindle speeds $R$ [r/min] | Opening time $t_0$ [μs] | |
| 1000 | 411.7 | 2256.7 |
| 4500 | 91.5 | 501.5 |
| 7000 | 58.8 | 322.4 |

**Table 2.** Experimental conditions for measuring temperature on blackbody furnace.

| | | |
|---|---|---|
| Outer diameter of grinding wheel | mm | 84 |
| Internal diameter of grinding wheel | mm | 64 |
| Optical holes on type A | | $\phi$10 mm × 4 |
| Optical holes on type B | | $\phi$18 mm × 2 and oval holes with a major axis of 18 mm |
| Interval holes | deg. | 90 |
| Measurement distance | mm | 100 |
| Temperature of blackbody furnace | °C | 300 |
| Measurement wavelength | μm | 2.0~6.8 |
| Response time of infrared thermometer | μs | 100 |
| Temperature measurement target size | mm | $\phi$8 |
| Sampling rate of temperature measurement | kHz | 50 |
| Spindle speeds | r/min | 1000, 4500, 7000 |

### 3.2. Experimental Results

Figure 6 presents the measurement results for various spindle speeds and grinding wheel combinations. Yellow areas of the graphs indicate the output result when the temperature measurement area of the infrared thermometer was located in the blackbody furnace, and gray areas are the output result when it was on the surface of the grinding wheel. Note that the time axis is different for each case. At 1000 r/min, the top of the temperature waveform was almost the same as the temperature of the blackbody furnace (300 °C), because both grinding wheels had a $t_0$ margin for a response time of 100 μs. At 4500 r/min, in the case of Type A, the $t_0$ was 91.5 μs, slightly less than the response time of 100 μs, so the result was about 2 °C lower than the temperature of the blackbody furnace. However, this was considered an acceptable error. On the other hand, since the $t_0$ of Type B was 322.4 μs, and there was a margin for the response time, approximately 300 °C was measured at the top of the waveform. Finally, at 7000 r/min, for Type A, the $t_0$ was 58.8 μs, shorter than the response time, so slightly below 300 °C, but the difference was about 5 °C, which again was considered an acceptable error for this kind of temperature measurement. For type B, almost 300 °C could be measured because the $t_0$ was increased.

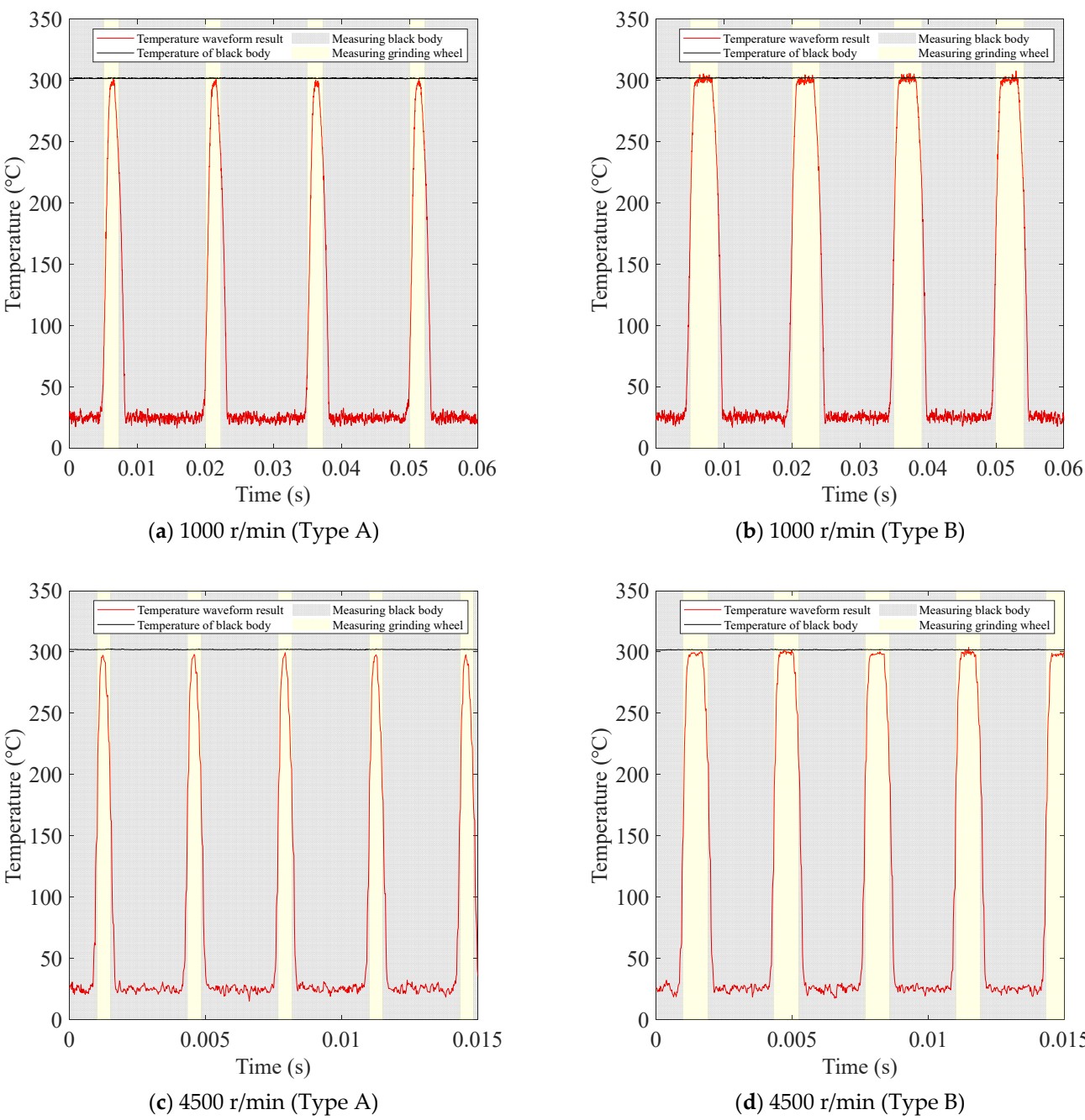

(**a**) 1000 r/min (Type A)

(**b**) 1000 r/min (Type B)

(**c**) 4500 r/min (Type A)

(**d**) 4500 r/min (Type B)

**Figure 6.** *Cont.*

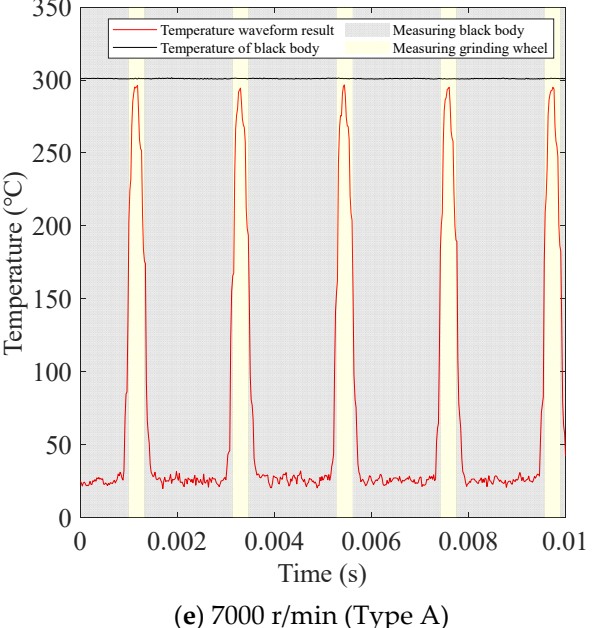

(**e**) 7000 r/min (Type A)

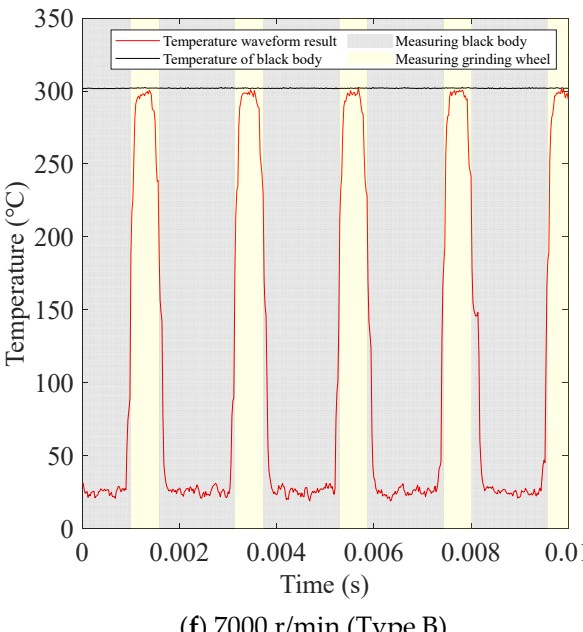

(**f**) 7000 r/min (Type B)

**Figure 6.** Temperature measurement result on blackbody furnace with different configurations of grinding wheel.

## 4. Application of Temperature Measurement during Dry Grinding of CFRP

### 4.1. Experimental Conditions

Next, we conducted experiments to demonstrate the effectiveness of the proposed temperature measurement method. We used it to measure the grinding surface temperature during the dry grinding of CFRP. The mechanical properties of CFRP may be reduced at temperatures higher than the glass transition temperature of the epoxy resin. It is therefore necessary to pay attention to the temperature during the grinding process. In addition, dry machining is often applied to avoid moisture penetration.

Table 3 lists the mechanical properties of the CFRP used in the experiment. This CFRP was a quasi-isotropic carbon/epoxy laminate in which unidirectional carbon fiber prepregs (TOHO TENAX QU135-197A) were laminated in combinations of 0, −45, 45, and 90 deg. The thickness of the CFRP plate was 7.8 mm, and two CFRP plates were stacked to make the measurement area larger than 8 mm of the target size of the infrared thermometer.

**Table 3.** Specifications of CFRP plate.

| Carbon fiber | | TOHO TENAX QU 135-197A |
|---|---|---|
| Resin | | Epoxy resin #135 |
| Fabric weight | g/m$^2$ | 190 |
| Curing temperature | °C | 180 |
| Thickness of prepreg Number of layers | mm | 0.187 40 |

Figure 7 and Table 4 present the experimental setup and grinding conditions, respectively. Instead of the blackbody furnace, the grinding surface of the CFRP plate was measured. The grinding surface temperature during end-face grinding of the CFRP was measured at cut depths of 0.02, 0.2, and 1.0 mm, assuming precision grinding, rough grinding, and high-efficiency grinding, respectively. An infrared thermometer was installed so that the optical path of the sensor was perpendicular to the tool axis, and the temperature measuring area was located at the center of the small hole on the workpiece side. The emissivity of CFRP was set to 0.952 by a calibration test. The color of the machined surface

of the CFRP does not change during machining. It is different from that in metal grinding, as metals are often oxidized. Thus, the effect on the emissivity change of CFRP is considered to be small.

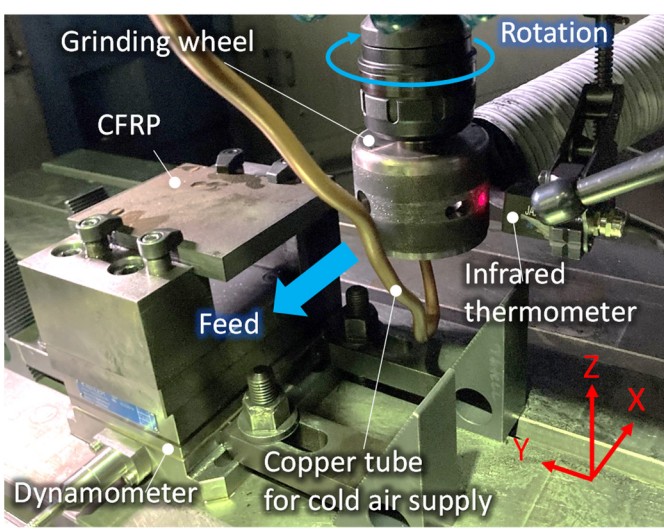

**Figure 7.** Experimental setup.

**Table 4.** Grinding conditions.

| Grinding wheel | | Type B |
|---|---|---|
| Optical holes on type B | | $\Phi$18 mm $\times$ 2 and oval holes |
| | | with a major axis of 18 mm |
| Rotation speed | r/min | 4500 |
| Grinding speed | m/s | 19.8 |
| Feed rate | mm/min | 1000 |
| Depth of cut | mm | 0.02, 0.2, 1.0 |
| Grinding direction | | Down cut |
| Grinding condition | | Internal cold air supply |
| Emissivity of CFRP | | 0.952 |
| Temperature measurement area position | | Center |
| Air volume | L/min | 464 |
| Air temperature | °C | 13 |
| Sampling rate | kHz | 50 |

In CFRP dry grinding, chips are scattered, and there was a concern that some could block the optical path for the measurement process. Therefore, as shown in Figure 8, cold air was supplied from the center of the cover plate beneath the HGW to prevent chips from falling into the wheel. The cold air was also expected to have a cooling effect on the grinding surface. Rodríguez. A et al. [25] revealed that delamination was suppressed by using $CO_2$-cryogenic cooling when drilling CFRP-Ti6Al4V stacks. This paper suggests that temperature control is important when machining CFRP.

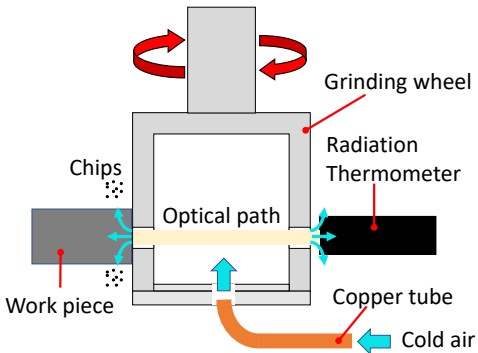

**Figure 8.** Cold air supply to prevent the effect of chips.

### 4.2. Experimental Results

Figure 9 shows the measurement results. The yellow areas of the graphs are the output result when the temperature-measuring area of the infrared thermometer was on the grinding surface, and the gray areas are the output result from the surface of the grinding wheel. It can be seen that the temperature of the grinding surface was measured periodically when the optical path opened every 1/4 rotation of the grinding wheel at any depth of cut. Note that the output value when the temperature-measuring area was on the surface of the grinding wheel is the value corrected by the emissivity of CFRP, and so is different from the true temperature of the surface. Figure 10 contains images of the emissivity of the surface of the grinding wheel after cleaning (a), with some clogging with a removal volume of 3146 mm$^3$ (depth of cut = 0.02 mm × 19 path, 0.1 mm × 18 path, 0.5 mm × 5 path, 1.0 mm × 20 path, 1.5 mm × 5 path, grinding distance 100 mm) (b), and wholly clogged with CFRP chips with a removal volume of 5722 mm$^3$ (depth of cut = 0.02 mm × 6 path, 0.1 mm × 16 path, 0.5 mm × 5 path, 1.0 mm × 40 path, 1.5 mm × 1 path, 2.0 mm × 4 path, 3.0 mm × 1 path, grinding distance 100 mm). Since the emissivity of the grinding surface increases as it is clogged with CFRP chips, it is expected to be between 0.79–0.88 during machining. Therefore, at depth of cut of 1.0 mm in Figure 9c, when measuring the temperature of the surface of the grinding wheel, 30 °C is 33–37 °C when converted to its emissivity. In the proposed measurement with infrared thermometer, the average temperature within a circle of ϕ 8 mm is measured and the spatial resolution is limited. We think that it is sufficient to set the average emissivity for the purpose of monitoring the temperature under actual grinding. From the above, we confirmed that the proposed temperature measurement method using an infrared thermometer could measure the temperature of the grinding surface and that of the grinding wheel in-process during the dry grinding of CFRP at any depth of cut, assuming precision grinding, rough grinding, and high-efficiency grinding.

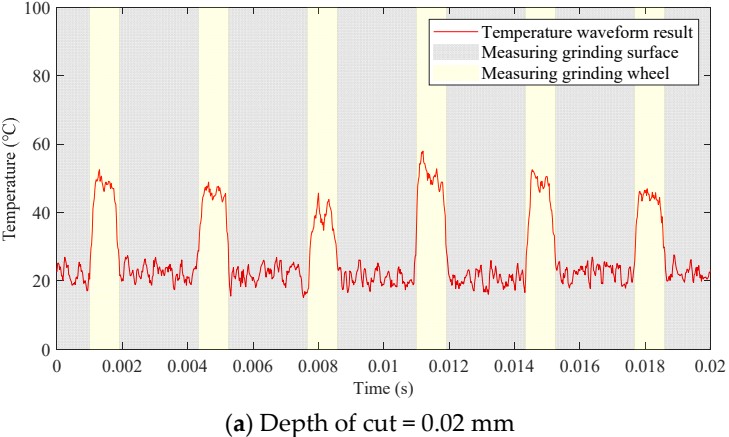

(**a**) Depth of cut = 0.02 mm

**Figure 9.** *Cont.*

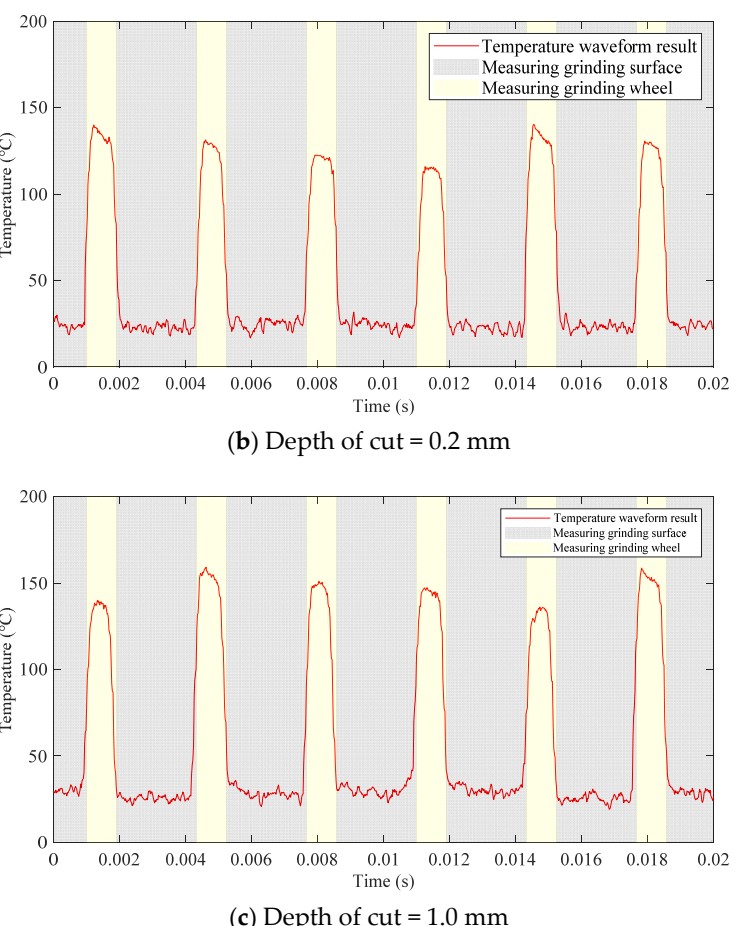

**(b)** Depth of cut = 0.2 mm

**(c)** Depth of cut = 1.0 mm

**Figure 9.** Transition of measured temperature on the machined surface and grinding wheel surface obtained from IR sensor, and the effect of depth of cut. (When measuring the temperature of the surface of the grinding wheel, 30 °C is 33–37 °C when converted to its emissivity.)

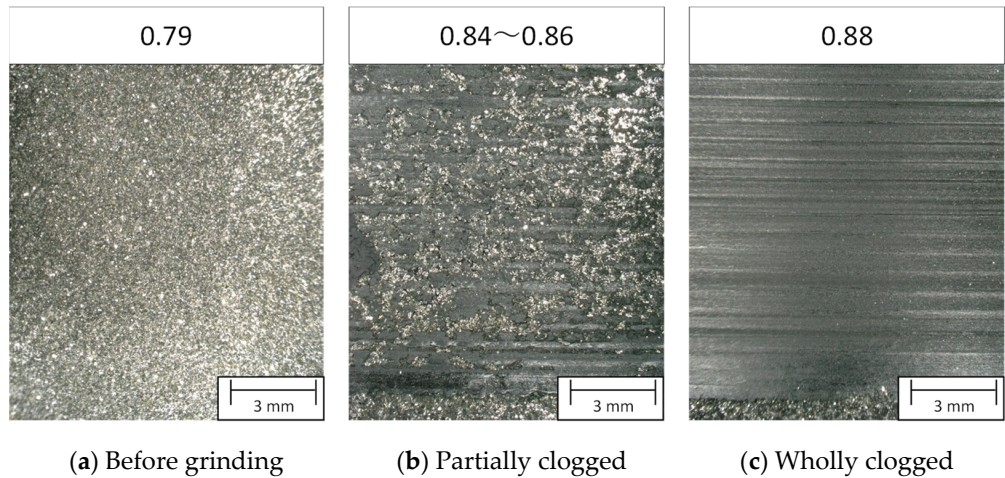

**(a)** Before grinding                  **(b)** Partially clogged                  **(c)** Wholly clogged

**Figure 10.** Transition of emissivity of grinding surface.

## 5. Relationship between the Position of the Temperature-Measurement Area of the Infrared Thermometer and the Measurement Value

### 5.1. Experimental Condition

Since the grinding surface has a temperature distribution, we predicted that the measurement value would change depending on the temperature measurement position of the infrared thermometer. Therefore, we investigated the effect of the position on the

measurement value. Table 5 lists the experimental conditions. Using a Type B grinding wheel, three patterns of temperature measurement position were tested: the finished surface side, the center, and the contact arc and pre-machined surface side (Figure 11). As an index of the temperature result of the proposed temperature measurement method, we used the "average peak temperature", which is the average value of the tops of the temperature waveform measured every 1/4 rotation of the grinding wheel during machining.

**Table 5.** Grinding conditions.

| Grinding wheel | | Type B |
|---|---|---|
| Rotation speed | r/min | 4500 |
| Grinding speed | m/s | 19.8 |
| Feed rate | mm/min | 1000 |
| Depth of cut | mm | 0.02, 0.2, 1.0 |
| Grinding direction | | Down cut |
| Emissivity of CFRP | | 0.952 |
| Temperature measurement area position | | Finished surface side, Center, Contact arc side |
| Air volume | L/min | 464 |
| Air temperature | °C | 13 |
| Sampling rate | kHz | 50 |

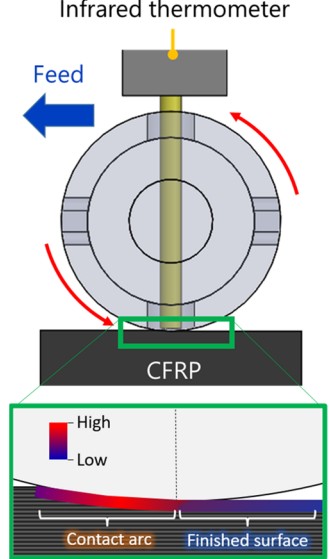

(**a**) Contact area between grinding wheel and CFRP

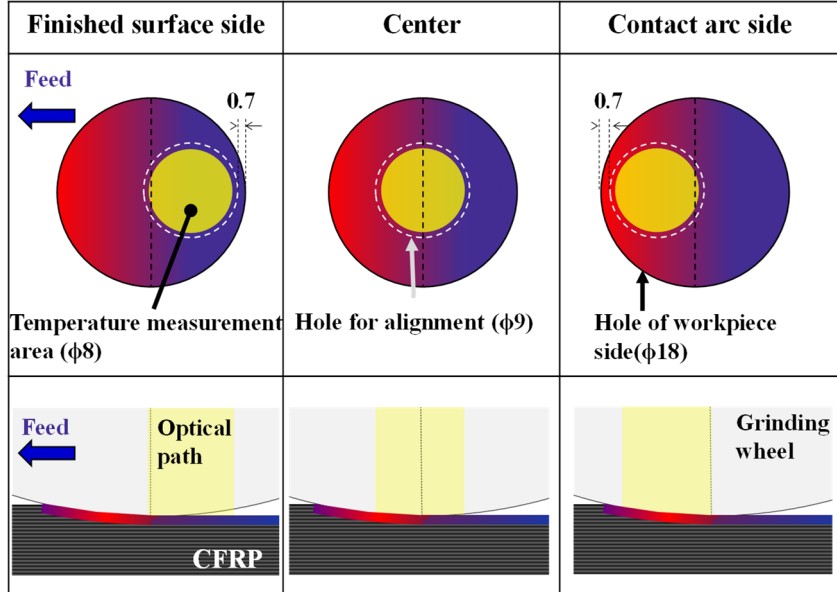

(**b**) Temperature measurement position on the grinding surface

**Figure 11.** Setting the position of the infrared thermometer.

### 5.2. Experimental Results

Figure 12 shows the temperature measurement results. At a depth of cut of 0.02 mm, the temperature was lowest when the measurement position was located on the contact arc side. At 0.2 mm, the temperature was highest when the position was the center. At 1.0 mm, the temperature was measured as high to low in the following order of contact arc side, center, and finished surface side. These results indicate that the magnitude relationship of the measurement temperature at each measurement position changes depending on the depth of cut.

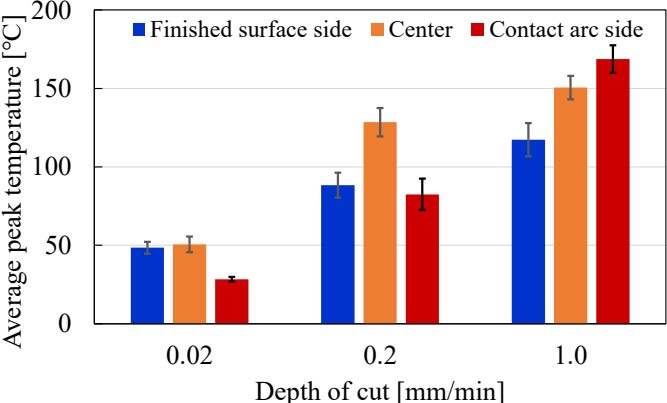

**Figure 12.** Relationship between the position of measurement area of infrared thermometer and measured average temperature.

*5.3. Numerical Analysis of Grinding Surface Temperature*

A quantitative evaluation of the relationship between the measurement value and temperature measurement position was performed. The analysis model is shown in Figure 13. In the analysis, it was assumed that the entire contact surface between the grinding wheel and workpiece was a band source of heat with uniform strength and a very large grinding width with respect to the contact arc length. Then, the temperature distribution on the surface of the workpiece when the heat source moved on the surface of the semi-infinite body at speed *v* was obtained. The actual shape of the contact surface was a curved surface along the outer circumference of the grinding wheel; however, since the depth of cut was sufficiently smaller than its diameter, it was regarded as a rectangle consisting of the grinding width *B* and the contact arc length 2*l*. Equation (2) shows the theoretical formula of the grinding surface temperature, improved by Takasawa et al. and based on the idea of the moving heat source of Jaeger in a previous study [26]. The analysis parameters are given in Table 6. The workpiece was isotropic material, and there was no heat conduction outside the contact area between the grinding wheel and workpiece. The temperature in the experimental environment and the horizontal grinding force were the same as the measurement values used for the machining experiment in the analysis, performed under the same conditions. The specific heat and thermal conductivity of CFRP were 0.68 J/Kg K and 3.17 W/m K, respectively, which were obtained in previous research [27]. In addition, a previous study showed that when dry grinding S45C with a WA grinding wheel at a depth of cut of 0.02 mm, the heat–flow ratio into the workpiece was about 70% and tended to decrease as the depth of cut increased [28]. Therefore, the heat–flow ratio was set to 70% when the depth of cut was 0.02 mm, 60% at 0.2 mm, and 50% at 1.0 mm.

$$\theta_{(z,x)} = \frac{2}{\pi} a \left( \frac{TV}{2JlB} \right) \cdot \frac{K}{k} \cdot \frac{1}{v} \int_{X-L}^{X+L} e^{-u} K_0 (u^2 + Z^2)^{1/2} du \qquad (2)$$

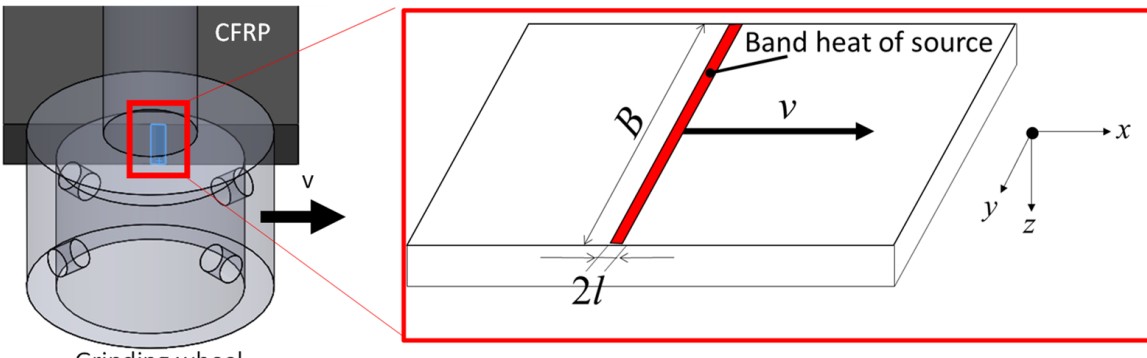

**Figure 13.** Heat source moving model for temperature analysis.

**Table 6.** Analysis parameters.

| | | |
|---|---|---|
| Environment temperature | °C | 20 |
| Grinding wheel diameter | mm | 84 |
| Depth of cut | mm | 0.02, 0.2, 1.0 |
| Flowing rate into workpiece $\alpha$ | % | 70, 60, 50 |
| Horizontal grinding force $T$ | N | 12.1, 35.0, 73.9 |
| Grinding speed $V$ | m/s | 19.8 |
| Equivalent of heat work $J$ | J/cal | 4.1855 |
| Contact arc length $2l$ | mm | 1.30, 4.10, 9.17 |
| Grinding width $B$ | mm | 15.6 |
| Thermal diffusivity of CFRP $K$ | J/kg·K | 0.68 |
| Thermal conductivity of CFRP $k$ | W/m·K | 3.17 |
| Feed rate $v$ | mm/min | 1000 |
| Modified Bessel function of the second kind of order zero $K_0$ | | |
| $X = vx/2K, L = vl/2K, Z = vz/2K$ | | |

### 5.4. Comparison of Experimental and Numerical Analysis Results

Figure 14 shows the temperature distribution of the grinding surface in the feed direction (*x*-direction) at each depth of cut. The solid black line represents the grinding surface. In addition, Figure 15 shows the two-dimensional (2-D) temperature distribution in the *x* and *y* directions at each temperature measurement position, assuming that the temperature distribution of the grinding surface in the feed direction is continuous in the plate thickness direction. The infrared thermometer outputs the average value of the temperature in the measurement target area. Here, we show the temperature distribution obtained by isolating the temperature measurement target area at each measurement position from the 2-D temperature distribution of the grinding surface obtained by numerical calculation. Figure 16 shows the results of comparing the average temperature obtained from this temperature distribution with the experimental results.

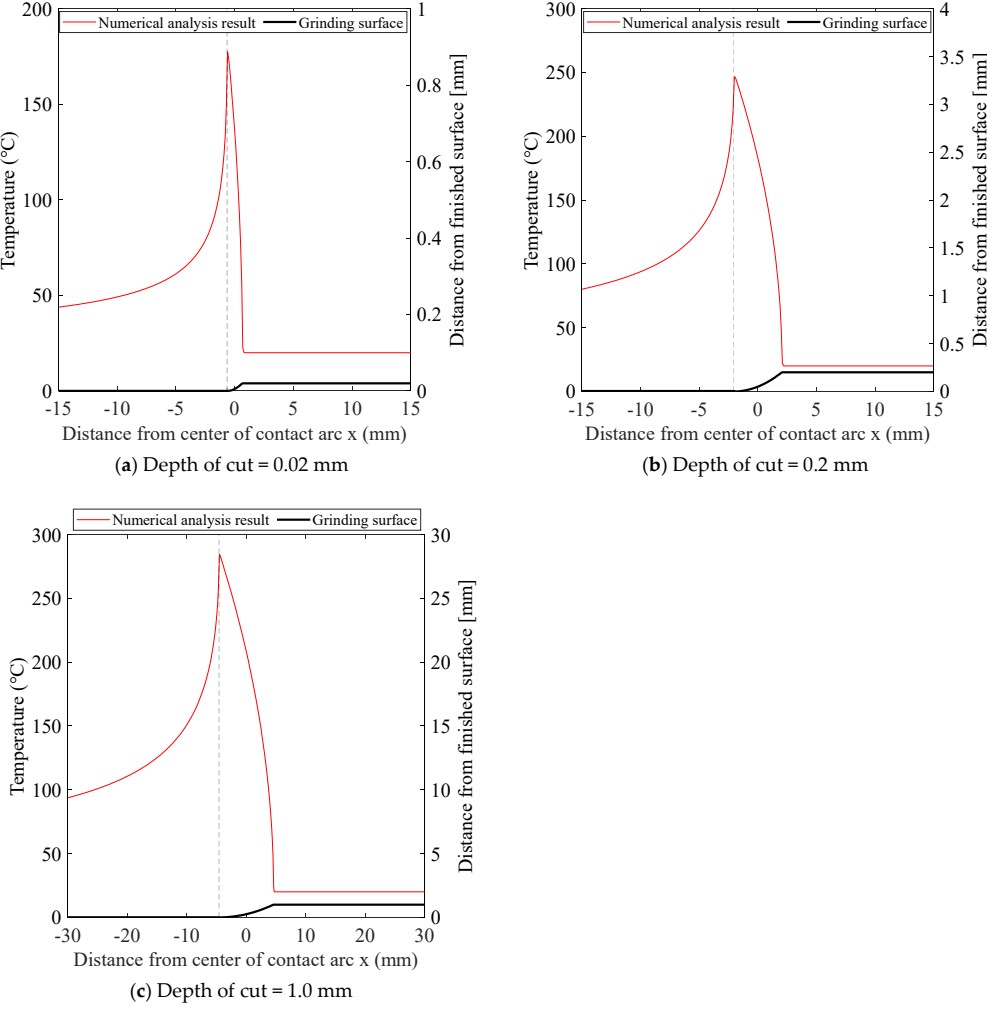

**Figure 14.** Temperature distribution in the feed direction.

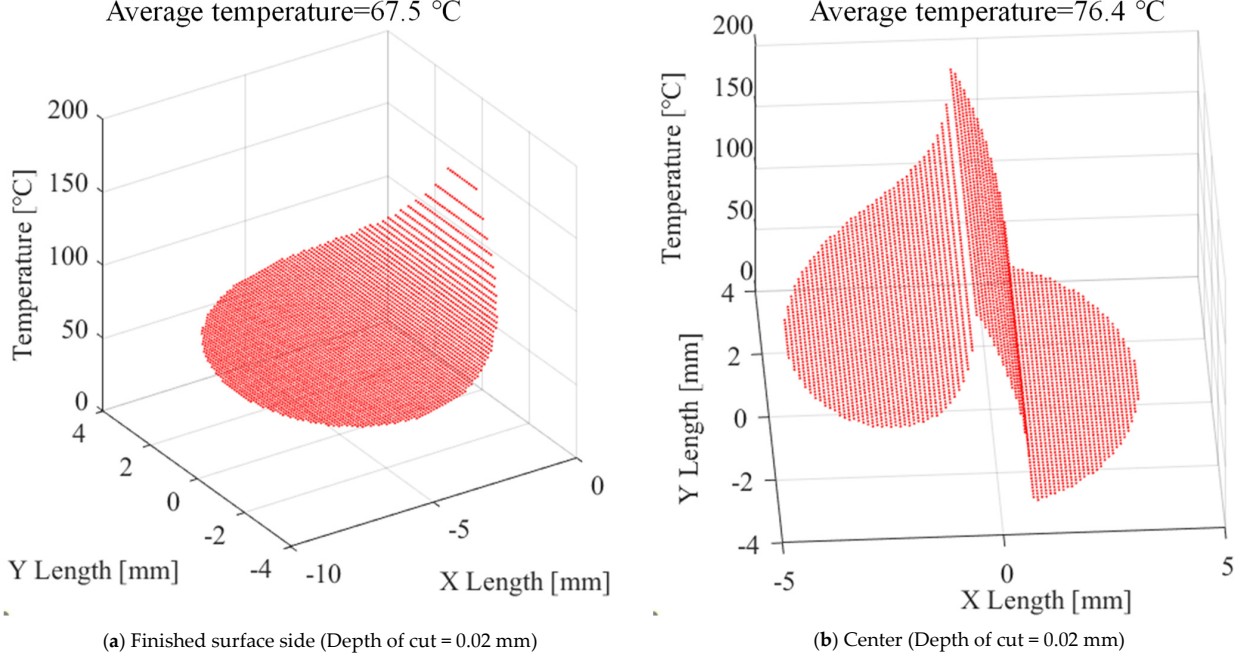

(**a**) Finished surface side (Depth of cut = 0.02 mm)

(**b**) Center (Depth of cut = 0.02 mm)

**Figure 15.** *Cont.*

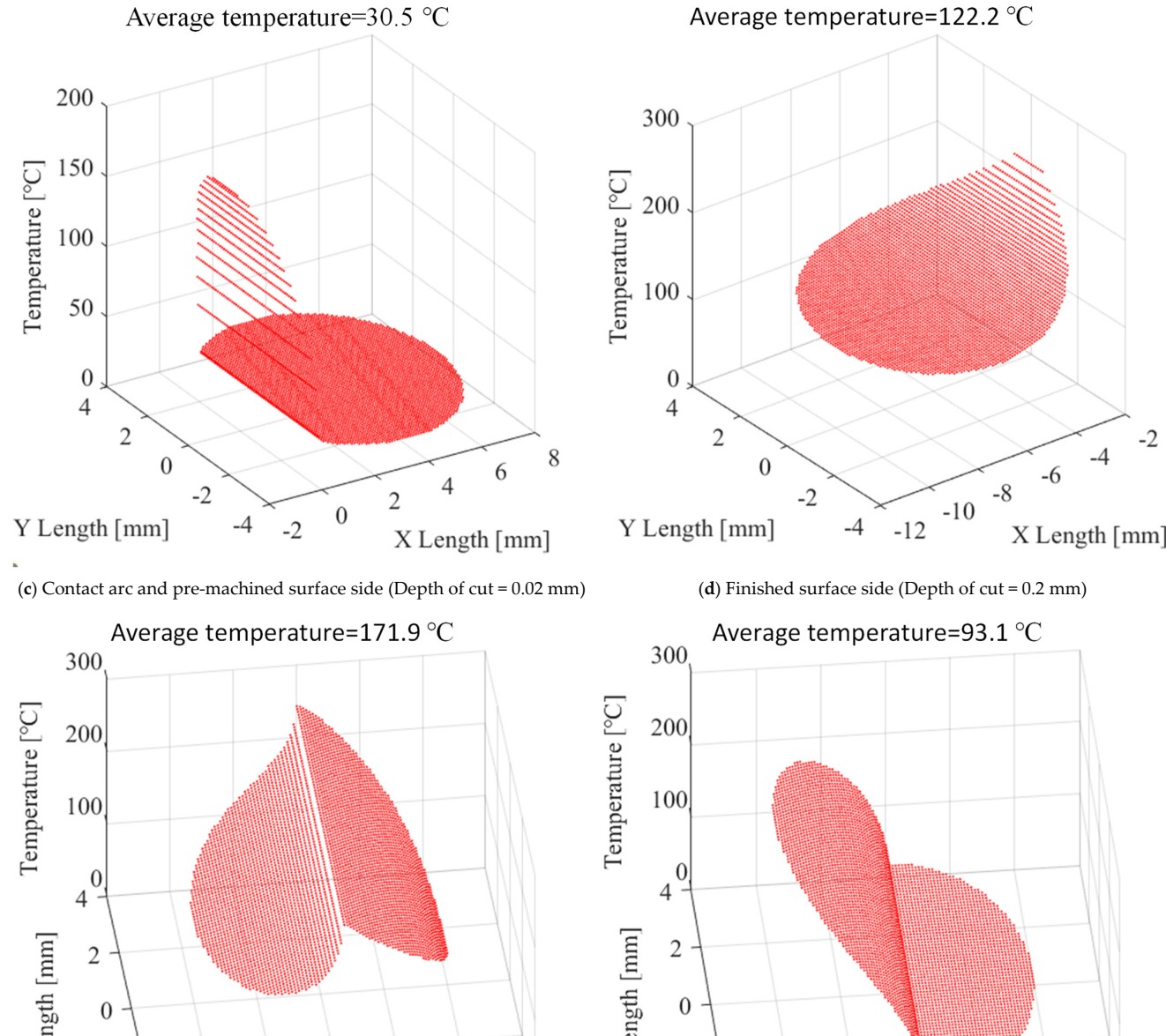

(**c**) Contact arc and pre-machined surface side (Depth of cut = 0.02 mm)

(**d**) Finished surface side (Depth of cut = 0.2 mm)

(**e**) Center (Depth of cut = 0.2 mm)

(**f**) Contact arc and pre-machined surface side (Depth of cut = 0.2 mm)

**Figure 15.** *Cont*.

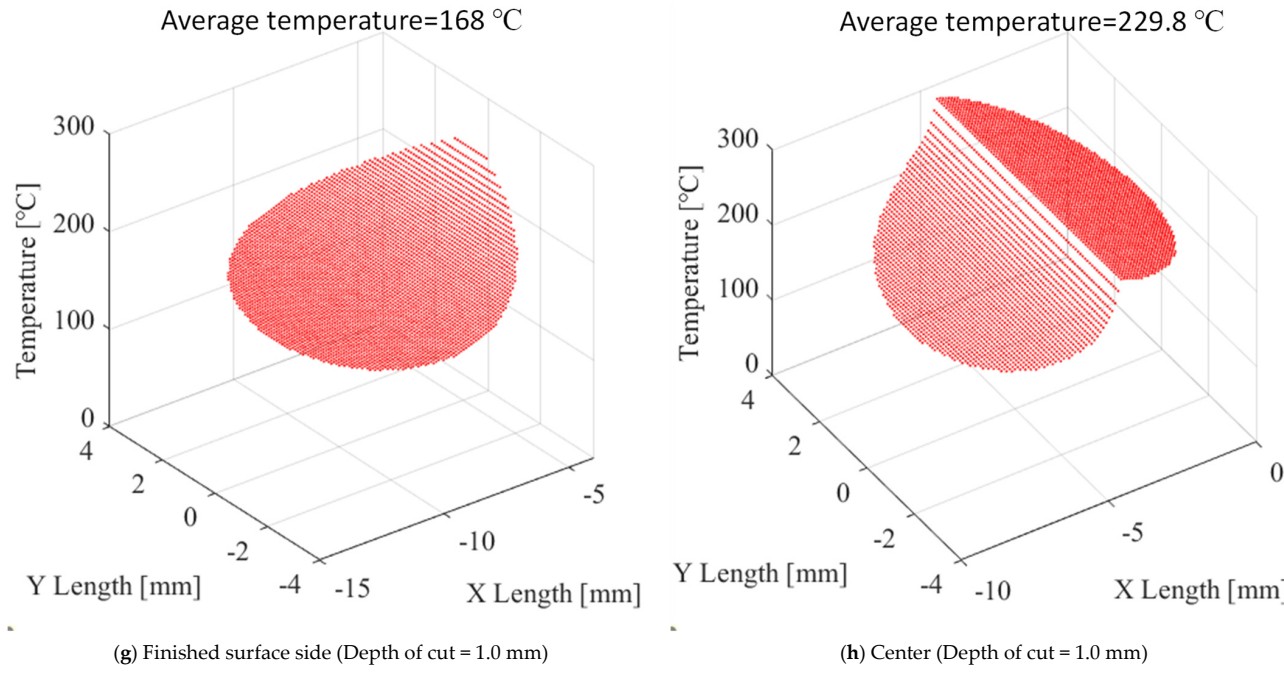

(**g**) Finished surface side (Depth of cut = 1.0 mm)   (**h**) Center (Depth of cut = 1.0 mm)

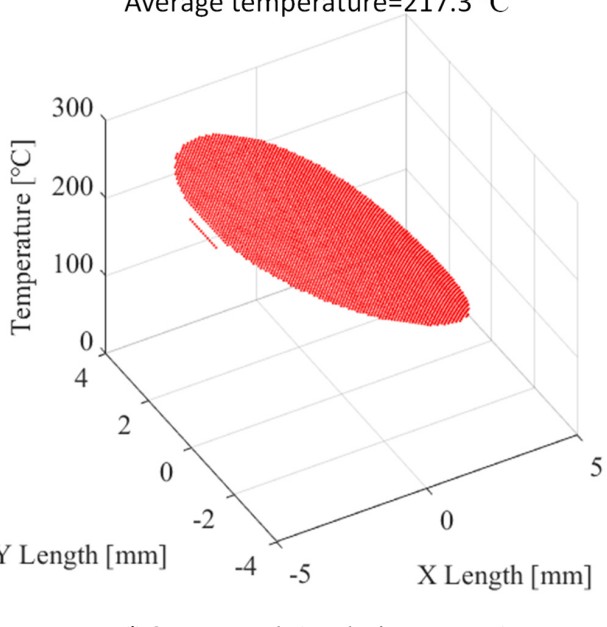

(**i**) Contact arc side (Depth of cut = 1.0 mm)

**Figure 15.** Two-dimensional temperature distribution at each temperature measurement position.

At a depth of cut of 0.02 mm, the tendency for the lowest temperature to be at the measurement position on the contact arc side was the same in both the experimental and numerical analysis results. In the experimental results, the average temperature at the measurement position on the contact arc side was 30.5 °C, which was 20 °C or lower than the other positions. This was because the contact arc length was 1.30 mm at a depth of cut of 0.02 mm, which is small with respect to the measurement target area diameter (ϕ8). Since the surface of the workpiece before machining occupied most of the measurement target area, the average peak temperature was lowest then. When the depth of cut was 0.2 mm, the tendency for the highest temperature to occur at the center measurement position was the same in both the experimental and numerical analysis results. This was because the temperature measurement position included the point where the temperature peaked and not that of the workpiece surface before machining at room temperature. Finally, at a depth

of cut of 1.0 mm, the tendency for the lowest temperature to occur when the measurement position was on the finished surface side was the same in both the experimental and numerical analysis results. On the other hand, the result showing that the temperature was higher in the center than on the contact arc side was different from the experimental result. It is considered that this was because when the depth of cut was 1.0 mm, the contact arc length was as large as 9.17 mm for a grinding width of 15.6 mm, which increased the difference from the heat source model assumed to move on a semi-infinite flat plate. Another possible reason is that as the contact arc length increased, the strength of the heat source became uneven and distributed, and the parameters of the heat source strength, such as the heat-flow ratio and specific grinding force, were different than when the depth of cut was as small as 0.02 mm.

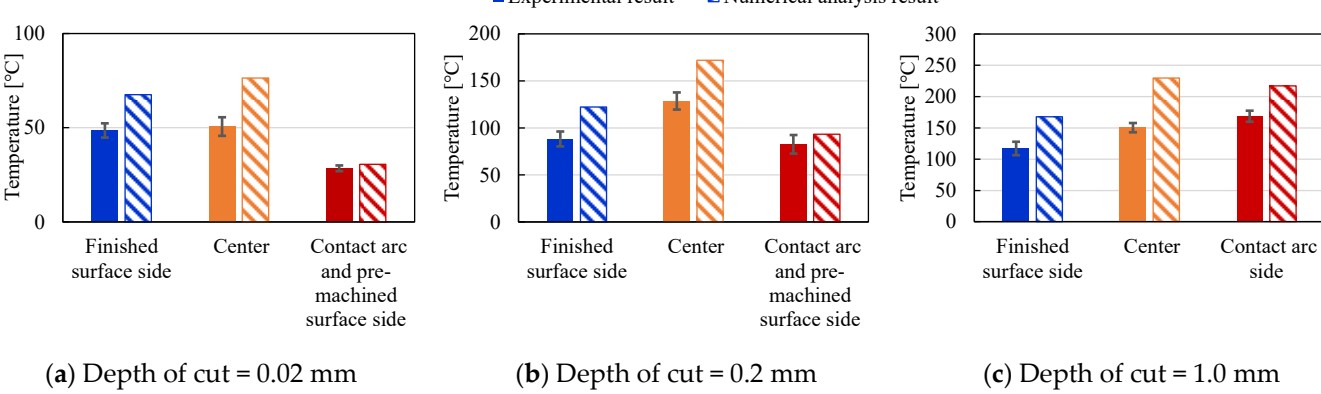

(**a**) Depth of cut = 0.02 mm  (**b**) Depth of cut = 0.2 mm  (**c**) Depth of cut = 1.0 mm

**Figure 16.** Comparison of temperature between experimental results and numerical calculation results.

Given these findings, since the temperature change on the grinding surface was steep, it was difficult to measure the maximum temperature with an infrared thermometer that measures the average temperature in the region of ϕ8. If the temperature measurement position was set to the contact arc side and center, the ratio of the region of the contact arc and the surface of the workpiece before machining to the entire temperature measuring area changed depending on the depth of cut. For example, in machining where the depth of cut is changed frequently, temperature changes at these measurement positions would be due to two factors: the change in the depth of cut and the evaluation target. On the other hand, on the finished surface side, the surface of the workpiece before machining at room temperature is not included in the temperature measurement area and the temperature in the range after the peak point is measured, so the same evaluation could be performed even if the depth of cut changes.

## 6. Application of the Detection of Machining Abnormalities

### 6.1. Experimental Conditions

We verified that the proposed temperature measurement method is effective in detecting abnormalities during grinding. Table 7 lists the experimental conditions. The grinding wheel used was Type A with a small hole diameter of ϕ10. CFRP end-face grinding was performed with a depth of cut of 1.0 mm and a grinding distance of 100 mm per path.

### 6.2. Experimental Results

In the 1 path, the machined surface was not damaged, and machining was performed normally. On the other hand, in the 2 path, the entire circumference of the grinding wheel was clogged due to the adhesion of CFRP chips (Figure 17), and fiber out occurred on the machined surface during processing (Figure 18). Figures 19 and 20 show the waveforms of the temperature and grinding force of the 1 path, where no abnormality occurred, and the 2 path, where an abnormality did occur, respectively. In the 1 path, there was no significant change in the temperature and grinding force; in the 2 path, in contrast, it can

be seen that the grinding force, grinding surface temperature, and temperature of the surface of the grinding wheel increased sharply, beginning at the machining midpoint. The temperature waveform around 0.9 s in the 2 path was similar to that in the 1 path. On the other hand, while the temperature waveform around 1.0 s was almost the same as that at around 0.9 s, a small peak appears for each rotation of the grinding wheel in the temperature waveform (blue circles and arrows in Figure 20c) when measuring the temperature of the grinding wheel surface. It is considered that this was because a part of the surface of the grinding wheel was clogged, and its emissivity increased, sharpness decreased, and temperature increased compared to other parts of the surface. By around 4.1 s, the temperature of the grinding surface and surface of grinding wheel rose to 450 °C and 200 °C or higher, respectively.

**Table 7.** Grinding conditions.

| Grinding wheel | | Type A |
|---|---|---|
| Optical holes on type A | | $\Phi 10 \times 4$ |
| Rotation speed | r/min | 2000 |
| Grinding speed | m/s | 8.8 |
| Feed rate | mm/min | 1000 |
| Depth of cut | mm | 1 |
| Grinding distance | mm | 100 mm × 2 path |
| Grinding direction | | Down cut |
| Grinding condition | | Internal cold air supply |
| Emissivity of CFRP | | 0.952 |
| Temperature measurement position | | Center |
| Air volume | L/min | 464 |
| Air temperature | °C | 13 |
| Sampling rate | kHz | 50 |

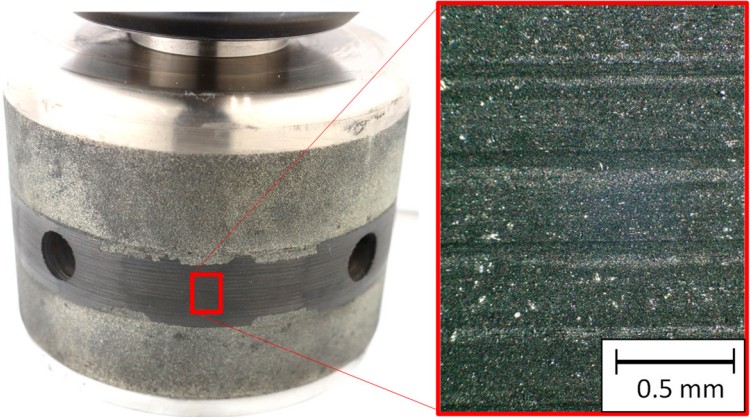

**Figure 17.** Grinding wheel after machining.

These results demonstrate that when the grinding wheel is clogged, both the temperature of the grinding surface and surface of the grinding wheel increase. The proposed temperature-measurement method using an infrared thermometer can detect such clogging from these two temperature increases. The grinding force also rises sharply when clogging occurs, but the change in the temperature waveform around 1.0 s, such as the occurrence of small peaks in the waveform when measuring the temperature of the surface of the grinding wheel, cannot be seen from the waveform of the grinding force. Therefore, it is

considered that the appearance of small peaks in the temperature waveform is superior to the change of the grinding force in detecting the clogging.

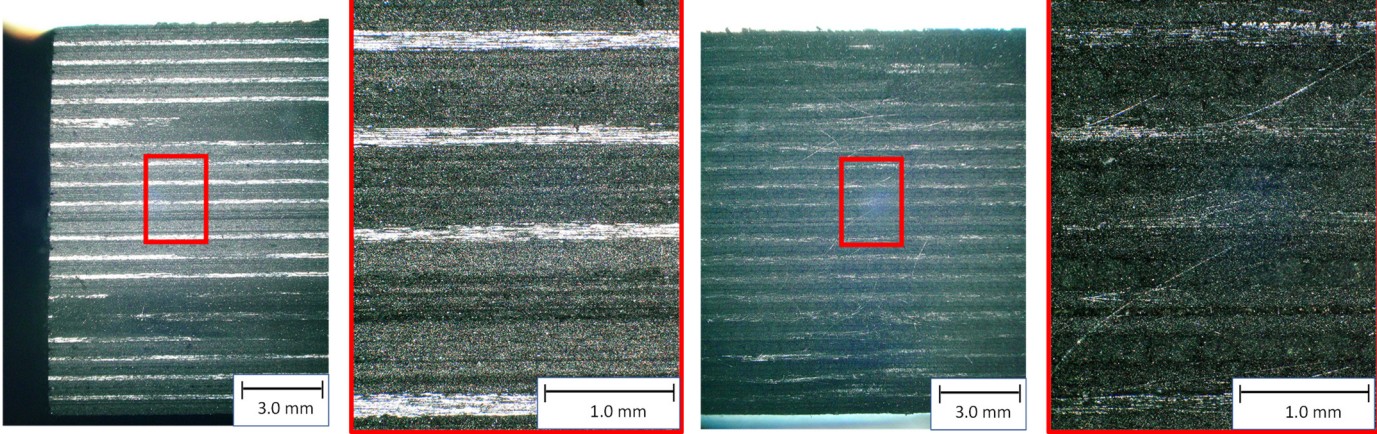

(**a**) Grinding start point        (**b**) 90 mm from grinding start point

**Figure 18.** Machined surface of CFRP.

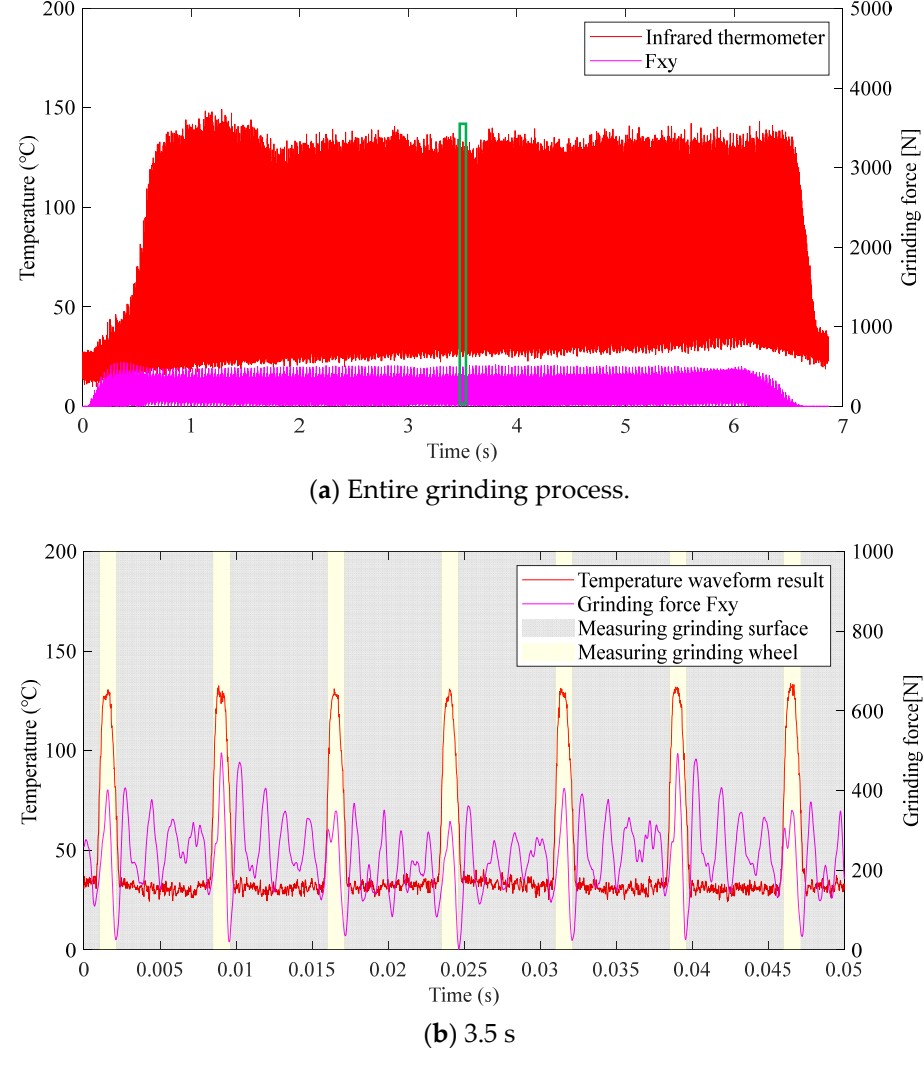

(**a**) Entire grinding process.

(**b**) 3.5 s

**Figure 19.** Transition of temperature and grinding force (1 path).

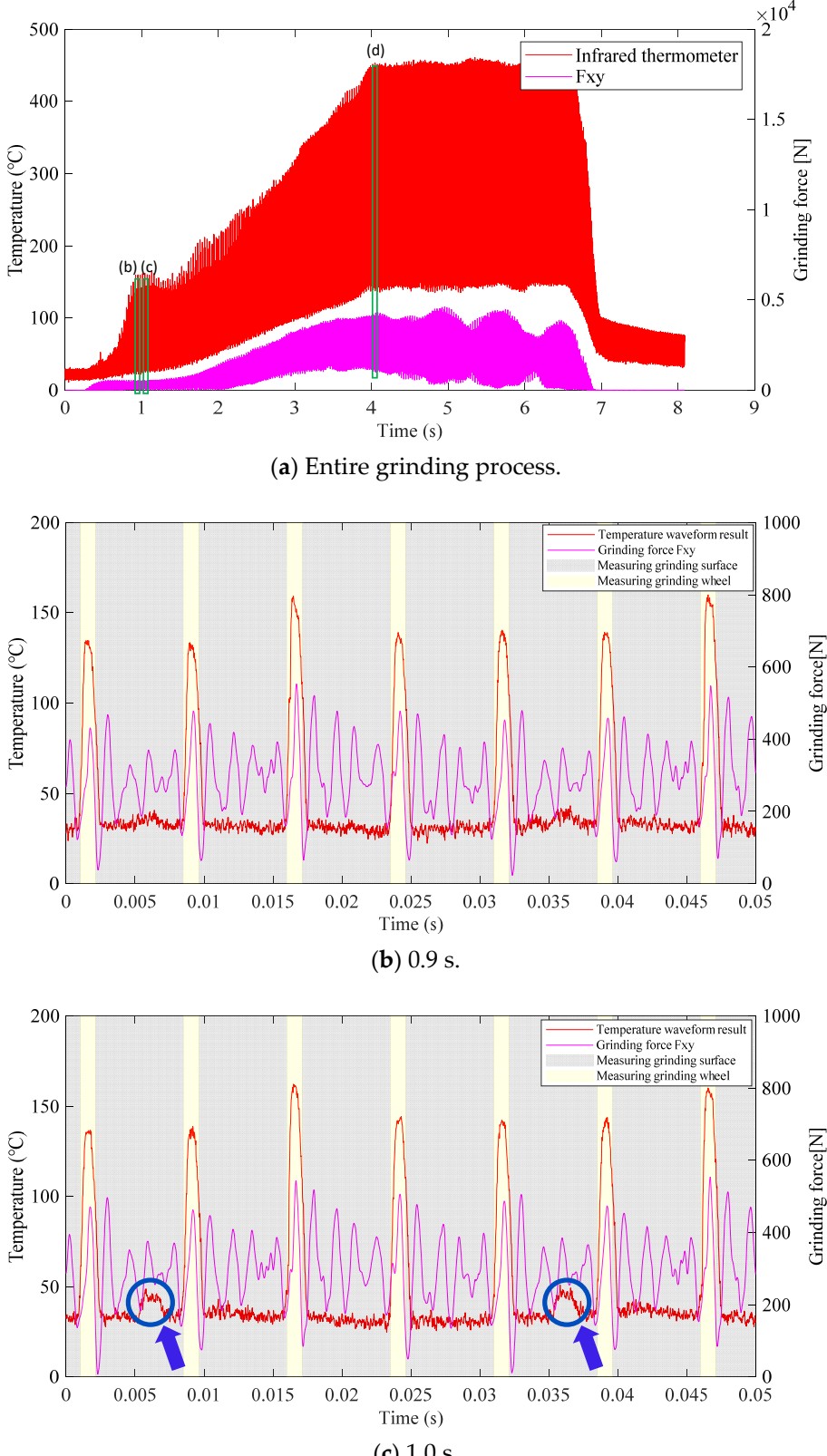

(**a**) Entire grinding process.

(**b**) 0.9 s.

(**c**) 1.0 s.

**Figure 20.** *Cont*.

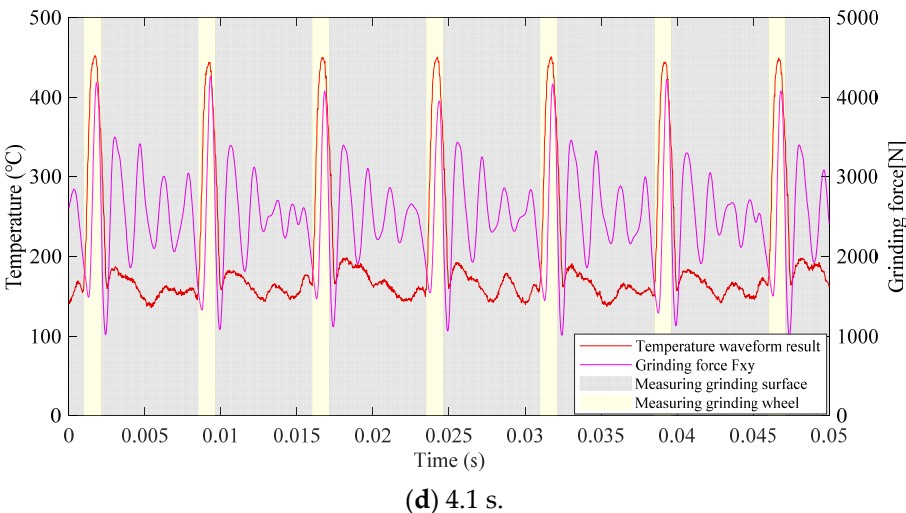

(**d**) 4.1 s.

**Figure 20.** Transition of temperature and grinding force (2 path).

## 7. Conclusions

In this study, we proposed a new temperature monitoring method for a machining grinding surface using an infrared thermometer and applied it to the dry grinding of CFRP.

(1) Small holes were created in a thin-wall hollow grinding wheel perpendicular to the tool axis, and the workpiece, grinding wheel, and infrared thermometer were arranged in that order. The temperature on the grinding surface could then be measured through the two holes on the grinding wheel. Alternatively, the temperature on the HGW could also be measured when the outer surface of the grinding wheel was placed within a temperature measuring area of the infrared thermometer.

(2) When the proposed method was applied to a blackbody furnace at 300 °C, it was possible to measure the temperature with an error of about 5 °C, even at a high-speed rotation of 7000 r/min.

(3) The temperature of the grinding surface and the surface of the grinding wheel could be measured in-process during the dry grinding of CFRP using the proposed temperature measurement method at any depth of cut, assuming precision grinding, rough grinding, and high-efficiency grinding.

(4) From the experimental and numerical analysis results, the measurement value changed depending on the temperature measurement position of the infrared thermometer. When the depth of cut was small (0.02 mm), the temperature, including the surface of the workpiece before machining, was measured if the position was on the contact arc side or center.

(5) When abnormal machining caused clogging while using the proposed method, a rapid temperature rise was observed in both the temperature of the grinding surface and the surface of the grinding wheel. Moreover, just before the rapid temperature rise occurred, there were small peaks in the waveform when measuring the temperature of the surface of the grinding wheel, which we consider could be an effective way to detect signs of clogging.

**Author Contributions:** Y.I. designed this study; Y.K. performed relevant experiments and wrote the manuscript, Y.F. advised; M.N. advised and manufactured the grinding wheel; H.S. supervised the experiments and data analysis and reviewed the manuscript. All authors have read and agreed to the published version of the manuscript.

**Funding:** This research received no external funding.

**Data Availability Statement:** Not applicable.

**Conflicts of Interest:** The authors have no conflict of interest to declare.

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
