# Peer review of "Development of In-Process Temperature Measurement of Grinding Surface with an Infrared Thermometer"

_jmmp, doi:10.3390/jmmp6020044_

Round 1

Reviewer 1 Report

  1. It is recommended to add up-to-date references in INTRODUCTION..
  2. Will the direction of cold air in Fig10 blow the chips towards the workpiece and cause adhesion? Would it be better to blow cold air along the feed direction of the grinding wheel?
  3. The size of the font in Fig12 is too large, it is recommended to adjust it according to the requirements of the journal.
  4. In Fig18, the deviation of experimental results compared to numerical analysis results can be calculated to make the comparison more specific.

Author Response

Thank you for your valuable comments.

The answers to the comments are as follows.

  • It is recommended to add up-to-date references in INTRODUCTION.

>Reference was added to the INTRODUCTION.

  • Will the direction of cold air in Fig10 blow the chips towards the workpiece and cause adhesion? Would it be better to blow cold air along the feed direction of the grinding wheel?

>The purpose of supplying cold air from the inside of the grinding wheel is to prevent chips from entering the optical path and adhering to the machined surface. When cold air is blown along the feed direction, chips may get inside the grinding wheel and cause measurement error.

  • The size of the font in Fig12 is too large, it is recommended to adjust it according to the requirements of the journal.

>The size of font was appropriately changed.

  • In Fig18, the deviation of experimental results compared to numerical analysis results can be calculated to make the comparison more specific.

>Thank you for your valuable comments. Experimental results show some kinds of error cause. It is rather difficult to compare specifically experimental results and analytical results.

Reviewer 2 Report

The idea is helpful, I see that results and approach can help many researchers working hard on grinding, many people mentioned the problem, some are referred by you, some are not, this point must be improved, because the definition of the problem must appeal to people in close techniques, such as SAM (super abrasive machining) Super abrasive machining of integral rotary components using grinding flank tools, Metals 8 (1), 24 gave early notice of the obstacles to get temperatures. Or in Precision Engineering 62, 204-212.

infrared thermometer: how did you calibrate the emissivity, the main aspect in Botlzmann equation? As known, bodies radiate energy W depending on temperature T, with the Stefan-Boltzmann law W = ε σT4 . Emissivity ε is equal to 1 for black bodies and less than 1 for grey bodies, σ being the Stefan constant. The emissivity of CFRP was set to 0.952 by calibration test  HOW?? The main aspect to discuss is it.

Figure 4 is very interesting.

This is the weaker point: emissivity description.

Othe rpeople working with milling/grinding of CFRP (grinding is not common) was defining many problems due to theal issues. Journal of Manufacturing Processes gave some new ideas in CFRP drilling with CO2 or LN2. The delamination was reduce dramatically.

SUMMARY: Good work, useful, but th emissivity aspects and complete the literatura are mandatory.

Author Response

Thank you for your valuable comments

We apologize for not providing a detailed description of the emissivity calibration method.

The emissivity calibration method in this study is to apply a blackbody spray to a part of the CFRP workpiece and heat it to about 100 °C. After that, measure the temperature of the part where the blackbody spray is applied with an infrared thermometer set to the emissivity of the blackbody spray (0.94). Then, measure the part without blackbody spray and adjust the emissivity setting so that the temperature of the part with blackbody spray is equal to the indicated value. The value obtained in this way was used as the emissivity of the workpiece.

Reviewer 3 Report

Temperature measurement of grinding surface has been always a difficult problem. This paper presents a new grinding temperature measurement method. The research methods and results are valuable. However, there are many questions need be improved and revised before it is published. Especially, the English language of this paper needs to be greatly revised. Some specific concerns/comments are listed below:

(1) The title needs to be revised to easily read for readers. The title should reflect the idea of the proposed new measurement method. The paper needs proofreading to improve the readability and clarity of the paper, especially the abstract, introduction and conclusion.

(2) The structure of the paper need to be properly adjusted. For example, the Figure 1, Figure 2 and Figure 4 should be merged, because they are all about experimental measurement devices. Figure 3 should be replaced in Introduction Section. Figures 9 and 4 are repeated.

(3) The vibration will occur in grinding process, especially in the motion of grinding wheel. How to ensure the stability of infrared temperature measurement? 

(4) In Abstract, CFRP needs its full name when it first appears.

(5) Some diagrams are not formatted properly. For example, Figures 12(a) and 12(b) are lack of scale. It is recommended to remove the shaded parts in the Figures, such as Figures 8, 11, 21, 22. The X and Y coordinates of Figures 16 and 17 are too small to be clear.

(6) In general, the grinding temperature increases as the depth of cut increases. Figure 18 shows that the experimental and numerical simulated temperatures of the grinding surface increase as the increase of the depth of cut from 0.02mm to 1.0mm. However, when the depth of cut is 0.02 and 0.2 mm, why are the temperatures of contact arc side smaller than the finished surface side?

Author Response

Thank you for your valuable comments.

The answers to the comments are as follows.

(1) The title needs to be revised to easily read for readers. The title should reflect the idea of the proposed new measurement method. The paper needs proofreading to improve the readability and clarity of the paper, especially the abstract, introduction and conclusion.

>The title is changed as follows and some of the text in the section you pointed out were revised.

Development of in-process temperature measurement of grinding surface

→Development of in-process temperature measurement of grinding surface and grinding wheel with an infrared thermometer.

(2) The structure of the paper need to be properly adjusted. For example, the Figure 1, Figure 2 and Figure 4 should be merged, because they are all about experimental measurement devices. Figure 3 should be replaced in Introduction Section. Figures 9 and 4 are repeated.

>Thank you for pointing out. The structure was modified as you pointed out.

(3) The vibration will occur in grinding process, especially in the motion of grinding wheel. How to ensure the stability of infrared temperature measurement?

>Under the grinding conditions of CFRP, grinding width 15.6 mm, and depth of cut 1 mm, there was no vibration problem. The measurements were always stable as shown in Figure 9(c).

(4) In Abstract, CFRP needs its full name when it first appears.

>Thank you for pointing out. The pointed-out part was corrected.

(5) Some diagrams are not formatted properly. For example, Figures 12(a) and 12(b) are lack of scale. It is recommended to remove the shaded parts in the Figures, such as Figures 8, 11, 21, 22. The X and Y coordinates of Figures 16 and 17 are too small to be clear.

>Thank you for pointing out. Scales were added to Figures 10 (a) and (b). In the Figures 6, 9, 19, 20, we set the shaded areas to help to understand the measured results more clearly. The yellow areas of the graphs are the output result when the temperature-measuring area of the infrared thermometer is on the grinding surface, and the gray areas are the output result from the surface of the grinding wheel. In the Figures 14, 15, the X and Y coordinates were increased. ※Due to the change in the structure, the figure number is different from the one in the comment.

(6) In general, the grinding temperature increases as the depth of cut increases. Figure 18 shows that the experimental and numerical simulated temperatures of the grinding surface increase as the increase of the depth of cut from 0.02mm to 1.0mm. However, when the depth of cut is 0.02 and 0.2 mm, why are the temperatures of contact arc side smaller than the finished surface side?

>Infrared thermometer measures the average temperature in the region of ϕ8. At a depth of cut of 0.02 mm,  contact arc side was significantly lower than the finished surface side, which measured the range after the peak point. This was because the area of the surface of the workpiece before machining occupies most of the measurement target area, the average peak temperature is then lowest on the contact arc side as shown Figure 15. Similarly, at a depth of cut of 0.2 mm, the effect of the area on the measured value could not be ignored, so the result was lower on the contact arc side than on the finished surface side.

Round 2

Reviewer 2 Report

Authors did not perform any of previous suggestions. They must read them all and provide a better version.

Summary:

  • Emissivity must be seriously discussed.
  • State of the art: they missed key Works, some even in MDPI about flank super abrasive machining and grinding
  • Ref 15 is old, people working in grinding provide recent works about monitoring: new sensors are in the markets, son even with 5G connection. Please give a serious discussion: follow authors recently publishing in the top Journal Mechanical systems and signal processing, or in Measurement: Urbikain, A del Olmo, Fei qin, etc… Grinding, machining is a key process, so an old state of the art is not a good thing in the recent times.

I kindly invite you to read all previous suggestions and provide a real better version.

Author Response

Thank you for your valuable comments.

We apologize for not being able to provide sufficient answers to comments in Round 1.

Firstly, the answers to the comments in Round 1 are as follows.

(Comments in Round1) The emissivity of CFRP was set to 0.952 by calibration test HOW?? The main aspect to discuss is it.

(Comments in Round2) Emissivity must be seriously discussed.

<Response>

The following three points were discussed regarding the emissivity of the grinding surface in the revised manuscript.

  1. Effect on emissivity due to change in color of the grinding surface: The color of the machined surface of the CFRP does not change during machining. It is different from that in metal grinding; metals are often oxidized. Then the effect on emissivity change of CFRP is considered to be small.
  2. Effect of chips in the optical path: The indicated temperature may decrease if the energy incident on the sensor is reduced by the chips existing in the optical path. To avoid this situation, cold air was supplied from the inside to the outside so that chips would not be present in the optical path.

3: Difference in emissivity between the grinding surface and the grinding wheel surface: The emissivity of the CFRP machined surface and the grinding wheel surface is significantly different. The emissivity of CFRP was 0.952, and the average emissivity of φ8 mm on the surface of the grinding wheel was about 0.79 to 0.88. This emissivity is taken into consideration when converting to the indicated temperature. In the proposed measurement with an infrared thermometer, the average temperature within a circle of Ï•8 mm is measured, and as shown in the manuscript, the spatial resolution is limited. We think that it is sufficient to set the average emissivity for the purpose of monitoring the temperature under actual grinding.

Comments:

Other people working with milling/grinding of CFRP (grinding is not common) was defining many problems due to thermal issues. Journal of Manufacturing Processes gave some new ideas in CFRP drilling with CO2 or LN2. The delamination was reduced dramatically.

The following reference was added to Section 4.1.

Rodríguez.A,; Calleja.A,; López de Lacalle. L.N,; Pereira.O,; Rubio-Mateos.A,; Rodríguez.G. Drilling of CFRP-Ti6Al4V stacks using CO2-cryogenic cooling, Journal of Manufacturing Processes, 2021, 64, 58-66

Comments:

State of the art: they missed key Works, some even in MDPI about flank super abrasive machining and grinding

Ref 15 is old, people working in grinding provide recent works about monitoring: new sensors are in the markets, son even with 5G connection. Please give a serious discussion: follow authors recently publishing in the top Journal Mechanical systems and signal processing, or in Measurement: Urbikain, A del Olmo, Fei qin, etc… Grinding, machining is a key process, so an old state of the art is not a good thing in the recent times.

Thank you for your comments. We removed Ref 15 and added six recent works, including MDPI and Mechanical systems and signal processing journals to the Introduction. They discuss the in-process monitoring capturing big data with multiple sensors, wireless transmitting the monitoring data, and a method utilizing machine learning of the acquired data to predict the process status, which is very important in the modern machining process.

Round 3

Reviewer 2 Report

Thank you for the new version. It is OK, and please go on with the interesting research based on this paper. The aspect were improved, and state of the art now is quite good.